# Engineering of Ti:Sapphire Lasers for Dermatology and Aesthetic Medicine

Aleksandr Tarasov *[ID] and Hong Chu

Laseroptek, 204 Hyundai I Valley, 31 Galmachi-ro 244 Beon-gil, Jungwon-gu, Seongnam 13212, Korea; hongchu@laseroptek.com
* Correspondence: aatarasov@laseroptek.com

**Abstract:** This review describes new engineering solutions for Ti:Sapphire lasers obtained at Laseroptek during the development of laser devices for dermatology and aesthetic medicine. The first device, PALLAS, produces 311 nm radiation by the third harmonic generation of a Ti:Sapphire laser, which possesses similar characteristics to excimer laser-based medical devices for skin treatments. In comparison to excimer lasers, Ti:Sapphire laser services are less expensive, which can save ~10% per year for customers compared to initial excimer laser costs. Here, the required characteristics were obtained due to the application of a new type of diffraction grating for spectral selection. The second device, HELIOS-4, based on the Ti:Sapphire laser, produces 300 mJ, 0.5 ns pulses at 785 nm for tattoo removal. The characteristics of HELIOS-4 exceed those of other tattoo removal laser devices represented in the medical market, despite a simple and inexpensive technical solution. The development of the last laser required the detailed study of a generation process and the investigation of the factors responsible for the synchronization of the generation in Ti:Sapphire lasers with short (several millimeters) cavities. The mechanism that can explain the synchronization in such lasers is suggested. Experiments for the confirmation of this concept are conducted and analyzed.

**Keywords:** solid-state lasers; Ti:Sapphire lasers; tunable lasers; subnanosecond lasers; induced scattering in crystals

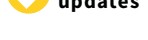



## 1. Introduction

The application of Ti:Sapphire lasers in medical practice is an exception rather than a rule. Only a few devices are represented in the medical laser market at this time. However, the unique properties of Ti:Sapphire crystals, such as their very wide luminescence spectrum, high emission cross section and figure of merit, in combination with excellent thermal characteristics, are highly favorable for the development of specialized medical laser devices with advantages and competitive ability against similar devices, realized on the basis of other active media.

Since 2012, our company, LASEROPTEK, has developed two medical laser devices, PALLAS and HELIOS-4, which include Ti:Sapphire lasers in combination with other modules, such as Nd:YAG lasers and nonlinear frequency converters. In this review, we omit a number of technical details and concentrate on new engineering solutions and on the results of laser physics studies obtained during the development of Ti:Sapphire lasers. Other related questions are described briefly to explain the actuality of the work and provide an overview of the devices.

The first device, PALLAS, was developed to generate UV radiation with the characteristics required for the light treatment of dermatology-related diseases, first of all, for psoriasis and vitiligo. The most efficient skin treatment here can be obtained using narrowband UV-B light (wavelength range 311–312 nm) [1]. Previous laser devices, used in practical dermatology since 2000, are based on 308 nm excimer XeCl lasers [2]. This wavelength is in close enough proximity to the UV-B range, and the high efficacy of ex-

cimer lasers with applications to skin treatment has been confirmed by numerous medical researchers [3].

In this regard, the serious disadvantage of the XeCl excimer laser is that the cost of consumables is very expensive. For instance, its after-sales support services include the frequent exchange of gas mixtures, cavity mirrors, windows and discharge electrodes, and it uses chlorine gas, which is highly reactive. Finally, annual service expenses costs more than 10% of the initial laser cost in Korea. From this point of view, the development of alternative lasers with low-cost consumables, based on solid-state lasers, is very important for users considering additional costs that can be saved.

The typical characteristics of excimer laser light pulses for skin treatment [3] include the following: pulse energy 3–6.5 mJ, pulse duration 10–50 ns, and pulse repetition rate 100–400 Hz. Such characteristics of XeCl laser were used as a base for PALLAS development. It is worth mentioning that the potential level of the XeCl laser's pulse energy is much larger, exceeding 100 mJ, but such high energy levels are not suitable for skin treatment.

It is possible to obtain the light with the required characteristics by the third harmonic generation (THG) of the Ti:Sapphire laser, operating at ~930 nm. The most advanced results in the generation of light at 300–315 nm on the basis of Ti:Sapphire lasers were obtained in 1999–2003 during the development of atmospheric ozone lidars [4,5]. They showed that the efficiency of THG by LBO crystals for ~1.0 MW pulses with nanosecond durations can exceed 30% if the laser generation is in single mode and the spectral width is less than 0.1 nm. To obtain the required spectral characteristics of the Ti:Sapphire laser, authors [4,5] used the injection seeding of narrow-line diode laser radiation in combination with preselection using a set of 4–5 dispersive prisms. It is also worth noting that well-known optical components, such as dispersion prisms, ruled metallic or holographic polymer diffraction gratings, and etalon Fabri-Perot, do not possess a total set of parameters, required for the successful realization of the abovementioned characteristics in Ti: Sapphire lasers.

During our work, we evaluated the application of new types of diffraction gratings: photo-thermo-refractive (PTR) glass volume Bragg gratings (VBG) [6] and fused silica subwavelength surface relief transmission gratings (STG) [7]. The original results of this study, which were obtained for the first time, were published in [8,9]. We review them in the following Section 2.1.

The second device, HELIOS-4, which includes the Ti:Sapphire laser, was developed for application in tattoo removal (TR) procedures. The popularity of laser TR is growing, and the present demand for lasers for TR is already significant. Preliminary studies in laser TR began in the mid-1960s and has been the most common method in clinical practice since the 1990s. Standard TR procedures at that time used Q-switched lasers with nanosecond pulse duration [10,11]. In 1998, it was shown that the usage of lasers with subnanosecond pulse duration considerably increases TR efficacy in comparison to treatment by nanosecond lasers [12]. The advantages of TR using the 795 nm radiation of the experimental Ti:Sapphire laser with a 500 ps pulse duration were demonstrated as far back as 1999 [13]. Therefore, it is surprising that the first TR device generating subnanosecond pulses, PICOSURE (from CYNOSURE), was developed only 16 years later, which obtained FDA approval in 2012. PICOSURE is based on the Cr:Alexandrite laser, operating at 755 nm with a pulse duration of 550–750 ps, maximum pulse energy of 200 mJ and pulse repetition rate of 10 Hz [14].

After 2012, several TR devices, based on Nd:YAG lasers with subnanosecond pulse duration, such as PICOWAY (CANDELA-SYNERON) and CUTERA (ENLIGHTEN), generating at 1064 and 532 nm, were developed and FDA approved. However, no other subnanosecond TR lasers generating radiation at 755–795 nm were developed until 2016.

The principle of subnanosecond pulse generation in PICOSURE is active mode-locking. It is notable that the Ti:Sapphire laser, used in [13], is operated by the same principle. In PICOSURE, mode-locking is combined with cavity dumping, which are both produced by an original electro-optical modulator with an original high-voltage fast electronic driver. To increase the active volume and obtain a relatively large (for mode-locked lasers) pulse energy, PICOSURE operates at several transversal modes. This leads to the essential deteri-

oration of laser stability, which is much worse than that of the Nd:YAG lasers mentioned above, based on more simple principles of operation. It is equally important to note that the disadvantage of PICOSURE is its high price, in particular, due to the high cost of an electro-optic system for mode-locking and cavity dumping. As a result, in 2012, the price of PICOSURE exceeded USD 200,000. Therefore, the development of simple and lower-cost subnanosecond TR lasers, which possess better stability and operate at 750–795 nm, also presented a problem.

In 2001, scientists from the Massachusetts Institute of Technology showed that single subnanosecond pulses can be generated in gain-switched Ti:Sapphire lasers with very short laser cavities [15] under pumping using pulses with a small enough duration. Pumping by the 532 nm radiation of Nd:YAG microlaser with a pulse duration of 600 ps, they obtained Ti:Sapphire laser generation at 775 nm with the energy of 18 µJ and pulse duration of 350 ps. The Ti:Sapphire laser cavity length was 3.5 mm. Following this approach, in 2016, CANDELA-SYNERON was developed and represented a new medical gain-switched subnanosecond Ti:Sapphire laser as an accessory for PICOWAY [16]. The Ti:Sapphire laser was made as a small cylinder, which can be embedded into the handpiece at the output of the PICOWAY articulated arm. It was pumped by 532 nm subnanosecond pulses of PICOWAY and generated light at 785 nm with a maximum energy of 130 mJ and pulse duration of 300 ps. Based on this, PICOWAY is a simpler and more economical solution than PICOSURE, but not in terms of maximum pulse energy. The possibility of a further increase of 785 nm in pulse energy in PICOWAY is restricted because the maximum energy of subnanosecond 532 nm pump pulses is 200 mJ. This energy does not exceed 300 mJ in any subnanosecond medical laser represented in the market.

A much larger pulse energy for pumping at 532 nm, without essential device complication, can be obtained in nanosecond Q-switched lasers. We investigated a gain-switched generation of subnanosecond pulses in short-cavity Ti:Sapphire lasers using pumping by nanosecond pulses. The original results were published in [17] and are considered in Section 2.2.

Section 2.3 considers the investigation of the processes, which leads to the synchronization of the generation within a large aperture in short-cavity Ti:Sapphire lasers. The synchronization plays an important role in the operation of gain-switched lasers, allowing the generation of subnanosecond pulses with 8–10 times shorter duration than the duration of pump pulses. The main results of this investigation have also been published before in [18,19].

## 2. Experimental Results and Discussion

### 2.1. Investigation of Ti:Sapphire Laser Operation with VBG and STG

VBG is holographic diffraction grating, which can operate as an output coupler of the laser cavity and produce back reflection at a given wavelength with a narrow spectral bandwidth. PTR glass, a VBG material, possesses improved thermo-optical characteristics and a larger laser-induced damage threshold (LIDT) than in polymer holographic gratings. STG is a wide class of diffraction gratings with deep subwavelength surface relief, which are produced by photolithographic technology with the etching of fused-silica substrates. In addition, some dielectric coatings can be deposited onto these substrates over surface relief to increase diffraction efficiency and make gratings polarization insensitive. Unfortunately, such coatings considerably reduce the LIDT of the component. We used TE-type STG without similar coatings (with only an AR coating on a flat rear surface, which has high LIDT). In addition to high LIDT, the spectral dispersion of such gratings is about two times larger than the dispersion of standard ruled gratings due to the smaller (subwavelength) groove period.

The parameters of the used gratings are listed in Table 1. We must clarify that the diffraction efficiency of our VBG, 54%, is made to correspond to the optimum reflection of the output coupler for the given operational conditions of the Ti:Sapphire laser. Passive losses in VBG, inserted into the laser cavity, are negligible. On the contrary, STG operates

in the Ti:Sapphire laser cavity as a bending mirror. This is why passive losses, inserted by FSTG and determined by radiation, transmitted into "0T"-order and reflected into "-1R"-order, are 100% − 95% = 5%. As a result, energy efficiency obtained for the Ti:Sapphire laser with VBG is larger than that for the laser with FSTG.

**Table 1.** Parameters of diffraction gratings.

| Name | STG | VBG |
|---|---|---|
| Operational diffraction order | -1T | 1R |
| Wavelength range | 830–1000 nm | Single wavelength, 933 nm |
| Angular dispersion | 2 mrad/nm | - |
| Spectral bandwidth | - | 0.15 nm |
| Diffraction efficiency | 95% | 54% (max > 99%) |
| Optimum angle of incidence | 44 deg | Normal incidence |
| Laser damage threshold (estimated) for 17 ns pulses | >30 J/cm$^2$ | <15 J/cm$^2$ |

The energy, spectral and spatial characteristics of the Ti:Sapphire laser were investigated with both types of gratings, with the laser operating at 50–100 Hz. Pumping of the Ti:Sapphire laser was conducted using the second harmonic of flash lamp-pumped electro-optically Q-switched Nd:YAG laser, which is capable of generating pulses with maximum $E_{1064}$ = 80 mJ at 100 Hz and ~95 mJ at 50 Hz. SHG was performed using a KTP crystal with an efficiency of more than 60%, so the maximum available 532 nm pump energy, $E_{PUMP}$, at 100 Hz was 53 mJ and 60 mJ at 50 Hz. For a correct comparison of the characteristics, all laser cavity parameters and pump spot sizes for the Ti:Sapphire crystal were made equal for both of the used gratings.

The dependence of output energy, $E_{OUT}$, at 932 nm on pump energy, $E_{PUMP}$, at 50 Hz is shown in Figure 1.

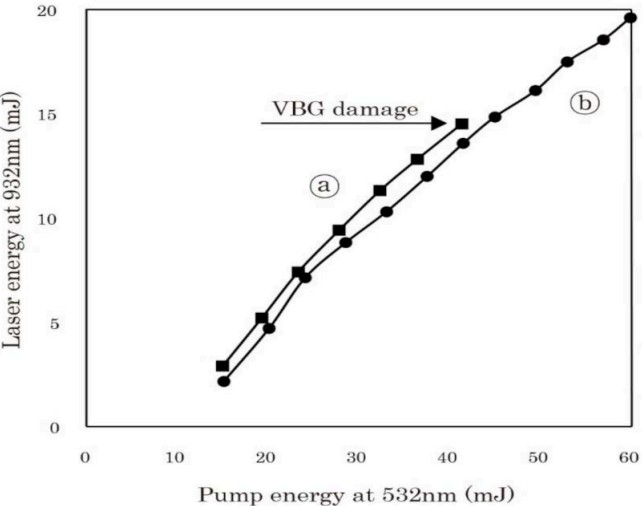

**Figure 1.** Energy characteristics of Ti:Sapphire laser with VBG (**a**) and STG (**b**). Reprinted with permission from ref. [8]. Copyright 2015 © The Optical Society.

The maximum laser efficiency obtained here with VBG was 35%, which is larger than the 32% obtained with STG. However, the maximum value of $E_{OUT}$ = 19 mJ for the FSTG laser was larger than the maximum $E_{OUT}$ = 14.5 mJ of the VBG laser. Energy restriction of the VBG laser was caused by VBG damage following a few seconds of operation at the indicated energy level. The estimated LIDT of these gratings did not exceed 15 J/cm$^2$. On the contrary, STG was not damaged during our study. The maximum $E_{OUT}$ in this case

was restricted by the maximum $E_{PUMP}$ = 60 mJ, which was 50 Hz for our pump source. Due to the high LIDT, we could expect the successful operation of STG at much larger energy densities than occurred here. The Ti:Sapphire laser energy's behavior at 100 Hz was identical to at 50 Hz, but the maximum $E_{OUT}$ of the STG laser was restricted here to 15 mJ by the reduction in pumping Nd:YAG laser energy (this is why the maximum $E_{PUMP}$ was reduced to 53 mJ). The following experiments were performed at 100 Hz.

Pulse duration of the STG laser at $E_{OUT}$ = 19 mJ was 17 ns, which increased to 30 ns at $E_{OUT}$ = 5 mJ. Therefore, the maximum pulse power here was 1.1 MW. Similar temporal characteristics were obtained for the VBG laser.

The spectral line width, $\Delta\lambda$, of the laser with STG, measured by a wavelength meter with a resolution limit ~30 pm at $\lambda$ ~ 1000 nm, was 33 +/− 2 pm at $E_{OUT}$ = 5 mJ, which increased to 48 +/− 2 pm at $E_{OUT}$ = 19 mJ. $\Delta\lambda$ at the VBG laser did not exceed the wavelength meter resolution limit at any level of $E_{OUT}$.

We could estimate the effective spectral resolution, $R_1$, of STG as the number of grating grooves illuminated by a laser beam. In our case, 1500 grooves/mm, a beam width of 0.7 mm, and an incidence angle of 44° obtained $R_1 \approx 1500$. The spectral resolution of VBG, $R_2 = \lambda/\Delta\lambda \approx 9000$, was much larger than $R_1$. The real value of the VBG laser spectral width is supposed less than 30 pm.

Wavelength deviation, $\delta\lambda$, for the laser with STG did not exceed 10 pm during ~5 min of operation at $E_{OUT}$ = 12 mJ. For the laser with VBG under the same conditions, we observed $\delta\lambda$ ~ 30 pm. We assume that the primary reason for wavelength deviation in the last case is VBG heating. According to [6], the thermal shift of VBG central wavelength, induced by PTR glass thermal expansion, is ~7 pm/K at $\lambda$ = 850 nm. At our wavelength, 932 nm, thermal shift was 7 pm/K × (932 nm/850 nm) $\approx$ 7.7 pm/K. Therefore, the observed $\delta\lambda$ corresponds to the VBG temperature increase <4 K, which may have been caused by room air heating. In comparison, the estimation of the STG wavelength shift induced by the thermal expansion of fused silica was ~0.5 pm/K, and only ~2 pm for a 4 K temperature increase. STG has a definite advantage over VBG for applications where laser generation with high wavelength stability is required. On the other hand, as mentioned in [6], the high sensitivity of the VBG central wavelength to temperature enables the fine tuning of the laser wavelength.

Wavelength tuning of the STG laser was produced by rotating the output mirror under a fixed incidence angle at grating and fixed pumping levels. The results are shown in Figure 2. The obtained tuning range was 170 nm. This range was restricted at wavelengths shorter than 840 nm by a reduction in the rear laser mirror reflection coefficient. It is worth mentioning that STG has a low wavelength dependence of diffraction efficiency and can be specially designed with reduced diffraction efficiency, but increased wavelength range.

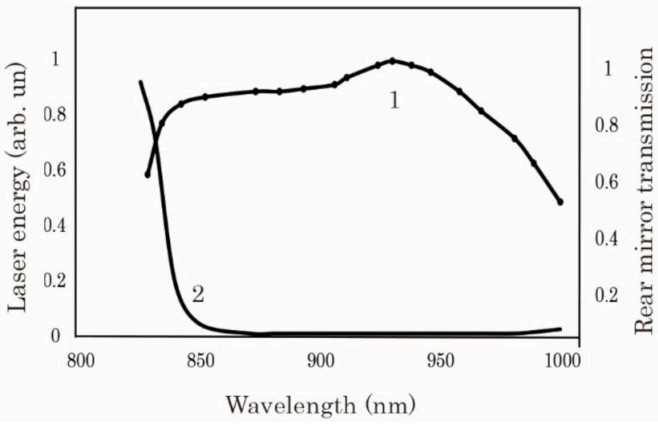

**Figure 2.** Wavelength dependence of Ti:Sapphire laser energy (1) and reflection coefficient of rear mirror (2). Reprinted with permission from ref. [8]. Copyright 2015 © The Optical Society.

THG was produced by two cascade LBO crystals: the first for SHG type I, and the second for THG type I, with a double wavelength wave plate after the first crystal to match 932 and 466 nm light polarizations for THG. Here, the efficiency of SHG was the same for both laser types, which exceeded 50% when $E_{932}$ > 12 mJ. The maximum THG efficiency of the VBG-type laser was 41%, close to the theoretically maximum value for the beams with a Gaussian profile. In the case of the STG laser, THG efficiency reached 36% at $E_{932}$ ~ 12 mJ and then, at larger $E_{932}$, reduced to 33%, restricting the growth of $E_{311}$ at ~5 mJ. We estimate that the main reason for THG worsening in the STG laser was the growth of the Ti:Sapphire laser spectral width when average pump power increased. Later, this problem was eliminated by optimization of the THG LBO crystal length. Nevertheless, here, the required $E_{311}$ at 100 Hz was realized using the Ti:Sapphire laser with both gratings.

We can conclude that the application of both types of gratings, in general, allows the required level of Ti:Sapphire laser characteristics to be obtained. Laser energy efficiency was larger with VBG, but the required $E_{932}$ here was obtained at very close to LIDT conditions. The maximum available $E_{932}$ was larger with STG due to the larger LIDT. Other characteristics were similar for both grating types. The main disadvantage of VBG for our application is the relatively low and unreproducible LIDT. Four VBG samples with the same specifications were tested, and all of them were damaged at $E_{OUT}$ > 12 . . . 14.5 mJ. On the contrary, consequent tests of 25 STG samples confirmed their high LIDT and reliability, so we rejected VBG in our devices in favor of STG.

Basing on these results, we developed the first solid-state dermatological UV laser PALLAS with a pulse energy of 5 mJ and repetition rate 100 Hz. The PALLAS laser head is shown in Figure 3. In 2017, LASEROPTEK started the serial production of these devices. In accordance with customer preferences, the laser wavelength can be set at 308 or 311 nm.

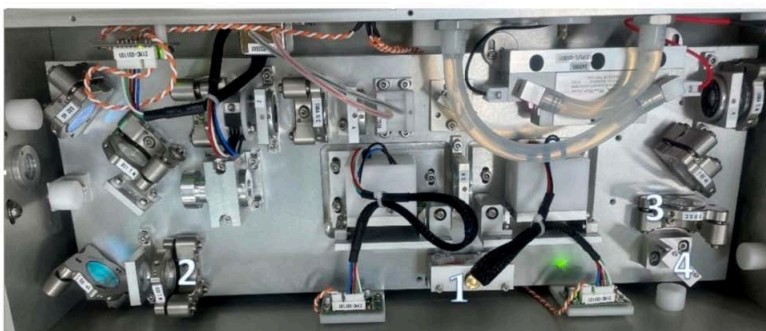

**Figure 3.** PALLAS laser head: 1 is Ti:Sapphire crystal (in the holder); 2 and 3 are total and output mirrors, respectively; 4 is STG.

*2.2. The Characteristics of Short Cavity Ti:Sapphire Gain-Switched Laser under Nanosecond Pumping*

To achieve the generation of single subnanosecond pulses by gain switching, the duration of pump pulse should be less than the build-up time of a generated pulse. Build-up time, $T_D$, and the duration of a generated pulse T reduce when pump fluence increases. When $T_D$ becomes less than the duration of the pump pulse, pumping will continue after the appearance of the first gain-switched laser pulse. The subsequent increase in pump fluence will lead to second pulse generation. This process restricts the maximum useable pump fluence and minimum gain-switched laser pulse duration. The theory in [20] predicts that the minimum duration of generated pulse can be several times shorter than the duration of pump pulse. In this study, for pumping, we used 532 nm radiation in our standard laser device, HELIOS-3. It consists of an electro-optically Q-switched multimode Nd:YAG master oscillator, followed by an amplifier and second harmonic generator. The maximum pulse energy at 532 nm here was 700 mJ with a pulse duration of 6 ns. We modified the resonator of the master oscillator and reduced the 532 nm pulse duration to 4 ns.

Using data [15], we could estimate the average value of pump fluence, $W_{PUMP} = E_{PUMP}/S_{PUMP}$, required to reduce the Ti:Sapphire laser pulse duration to below 1 ns: $W_{PUMP} \sim 1\ J/cm^2$. Here, $E_{PUMP}$ represents the total pump energy, and $S_{PUMP}$ is the pump spot area. Therefore, to obtain a laser pulse energy larger than that of PICOSURE, $E \geq 200$ mJ, the linear size of the square pump spot inside the Ti:Sapphire crystal had to exceed 4–5 mm.

An optical scheme of Ti:Sapphire laser is shown in Figure 4. Ti:Sapphire active crystals had an absorption coefficient of 3.5–4.0 cm$^{-1}$ at 532 and a thickness of 3 mm; they were AR-coated at 532 and 780–820 nm. The pump mirror had 96% transmission at 532 and 99.5% reflection within 780–820 nm. We used output mirrors with reflection from 40% to 75% to generate wavelengths. The output mirror had an additional high reflective coating at 532 nm to increase the absorption of pump radiation in the Ti:Sapphire crystal by reverse pass. Both cavity mirrors were flat. The distance between mirrors was 3.5–7.5 mm.

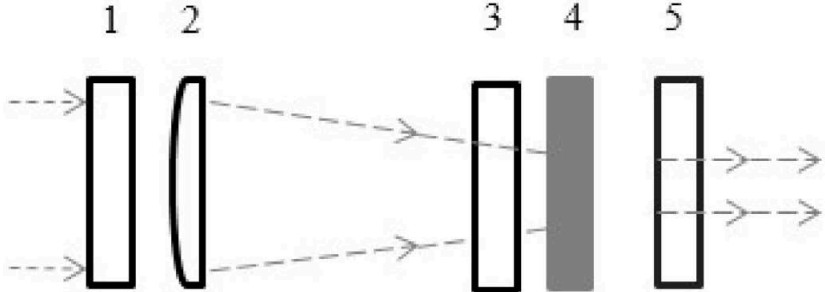

**Figure 4.** Optical scheme of the Ti:Sapphire laser. Here, 1 is microlens array homogenizer; 2 is the condenser lens; 3 and 5 are pump and output mirrors, respectively; and 4 is the Ti:Sapphire active crystal. Reprinted with permission from ref. [17]. Copyright 2019 © The Optical Society.

Laser pulse duration, T, was determined using a photodiode (UPD-50-UD, rise time <40 ps and fall time 50 ps, from Alphalas), in combination with a digital oscilloscope (Infiniium DSO 80804B, 8 GHz bandwidth, from Agilent). Pulse duration was determined for the light integrated from the whole laser aperture. For this purpose, a diffuser was placed at 50 cm from the Ti:Sapphire laser output mirror, and then the laser spot at the diffuser was imaged on the photodiode.

At the initial stage of the work, without homogenization, we directly focused 532 nm multimode pump radiation into a spot with the required size at the surface of the Ti:Sapphire crystal. The first results were disappointing: generated pulses have very unstable shape, varying from pulse to pulse, and the duration was larger than 900 ps (see Figure 5). Similar characteristics were obtained via pumping using a singlemode Nd:YAG laser. To explain these results, we proceeded from the assumption that a laser with a very short cavity and relatively large lateral size of active area (Fresnel number $N_F \geq 1000$), where no physical reasons exist for the coupling of a laser field in the lateral direction, can act as an array of multiple independent local channels with different lateral sizes and shapes and generate in parallel. In the case of flat mirrors, all channels have equal losses, but different gains, as determined by the local value of pump fluence. If the local gain is different, then the T and $T_D$ of the local channels will also be different. Therefore, the pulses from different channels will appear at different times. $T_D$ is several times larger than T, so the spreading of $T_D$ may considerably increase the duration of the signal integrated from all laser apertures. To minimize spreading, we must synchronize the generation of local channels by equalizing the $W_{PUMP}$ over laser aperture. With this purpose and for the elimination of possible damage to the Ti:Sapphire crystal by hot spots in the 532 nm beam, we arranged pump beam homogenization using two-dimensional microlens array (MLA) [21], as shown in Figure 4.

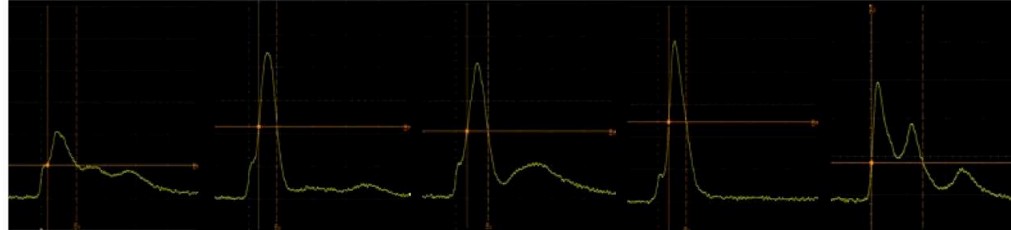

**Figure 5.** Series of Ti:Sapphire laser oscillograms, obtained without pump beam homogenization. Minimum duration of the main pulse is 930 ps. Reprinted with permission from ref. [18]. Copyright 2019 © The Optical Society.

After the homogenization, we obtained the stable generation of single pulses. Ti:Sapphire laser energy and temporal characteristics were measured for different combinations of pump spot size and cavity length. The dependences of the Ti:Sapphire laser pulse duration on $W_{PUMP}$ are shown in Figure 6.

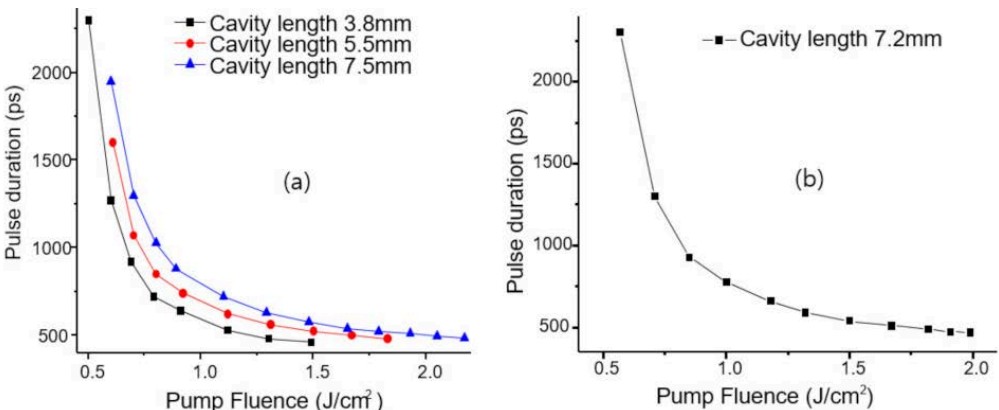

**Figure 6.** Dependences of the pulse duration T on pump fluence $W_{PUMP}$ for pump spot $3.4 \times 3.4$ mm (**a**) and $5.6 \times 5.6$ mm (**b**). Reprinted with permission from ref. [17]. Copyright 2019 © The Optical Society.

In general, the maximum useable $W_{PUMP}$ can be restricted by parasitic lasing inside the Ti:Sapphire crystal [22], by crystal damage under pumping or by the appearance of the second pulse. Each of these processes can appear depending on experimental conditions. For example, parasitic generation was observed in our experiments when the cavity length was greater than 13 mm. At cavity lengths shorter than 7.5 mm, parasitic generation was always suppressed by the generation in the main laser cavity due to a larger $T_D$. After pump homogenization, crystal damage appeared at $W_{PUMP}$, exceeding the threshold of the second pulse. All experimental data shown here were recorded within the range of single pulse generation. The last point of each curve corresponds to the threshold of the second pulse.

From Figure 6, one can see that when $W_{PUMP}$ exceeded ~1.4 J/cm$^2$, T remained almost unchanged. On the one hand, this did not allow us to achieve $T \leq 460$ ps. On the other hand, the low dependence of T on the $W_{PUMP}$ resulted in the high stability of T. At maximum values of $W_{PUMP}$, we obtained the standard deviation of T as low as 1%. The reduction in the cavity length from 7.5 to 3.8 mm did not allow us to reduce the minimum available T, and resulted only in a reduction in $W_{PUMP}$, which is required to achieve the minimum T. This can reduce the risk of laser crystal damage by setting a lower $W_{PUMP}$ without increasing T.

Figure 7 shows the dependence of the Ti:Sapphire laser pulse energy, E, on pump energy, $E_{PUMP}$, for two pump spot sizes. In both cases, (a) and (b), the maximum E was restricted by the second pulse appearance. To achieve a larger E, we had to increase both the $E_{PUMP}$ and pump spot size. In our case, the maximum pulse energy of 300 mJ at

T = 520 ps was obtained with a pump spot of 7.2 × 7.2 mm at the maximum available $E_{PUMP}$ of 700 mJ. Here, $W_{PUMP}$ did not exceed 1.35 J/cm$^2$, so the adjustment of the spot size allowed us to increase $W_{PUMP}$ and to achieve the same E with a shorter T.

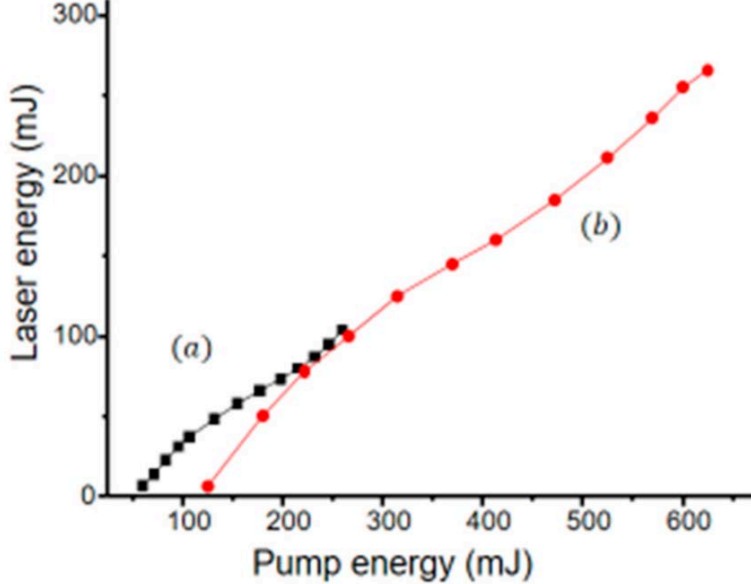

**Figure 7.** Dependence of E on $E_{PUMP}$: (**a**)—pump spot 3.4 × 3.4 mm, L = 7.5 mm; (**b**)—pump spot 5.6 × 5.6 mm, L = 7.2 mm. Reprinted with permission from ref. [17]. Copyright 2019 © The Optical Society.

The best values of Ti:Sapphire laser characteristics are shown in Table 2:

**Table 2.** Characteristics of Ti:Sapphire laser generation. Reprinted with permission from ref. [17]. Copyright 2019 © The Optical Society.

| 1 | Maximum pulse energy | mJ | 300 |
|---|---|---|---|
| 2 | Pulse energy stability | % | 2 |
| 3 | Minimum pulse duration | ps | 460 |
| 4 | Pulse duration stability | % | 1 |
| 5 | Maximum pulse repetition rate | Hz | 10 |
| 6 | Maximum energy efficiency | % | 43 |
| 7 | Central wavelength | nm | 785 |
| 8 | Spectral width | nm | <20 |
| 9 | Laser beam profile | | Flat-top |

Similar characteristics were obtained using another type of homogenizer—diffraction diffusers (DD) [23]. DD has diffraction efficiency ~75%, which led to a reduction in pump efficiency in comparison to MLA. Due to this, we selected MLA as a main component for homogenization in our laser devices.

We can conclude that subnanosecond pulses with durations of less than 500 ps and energy ≥300 mJ can be generated by the simple gain-switched Ti:Sapphire laser with a short (several millimeters) laser cavity when pumping is produced by a Q-switched laser with a pulse duration ~4 ns or less, with the appropriate homogenization of pumping spot. Based on these results, we developed a TR device, HELIOS-4, which combines the modified HELIOS-3 and the Ti:Sapphire laser module. A part of the HELIOS-4 laser head, with the Ti:Sapphire module, is shown in Figure 8. Pump radiation enters the module from the right side.

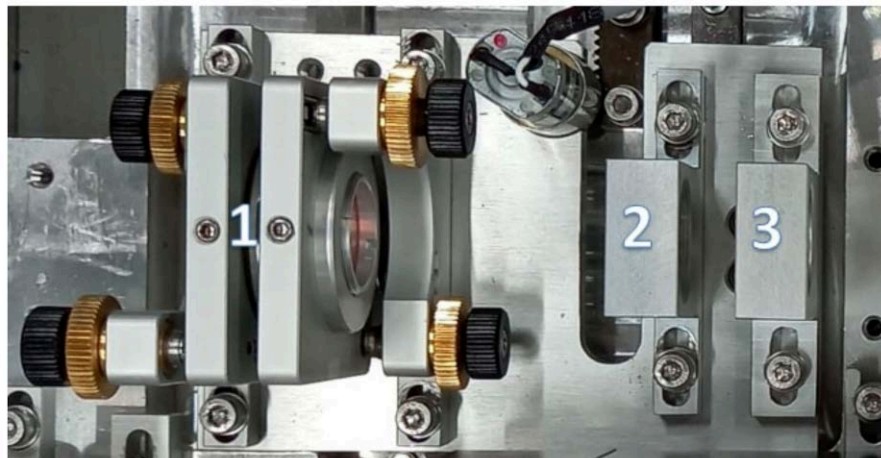

**Figure 8.** A part of HELIOS-4 laser head with assembled components for Ti:Sapphire laser: 1—mounts with Ti:Sapphire crystal and laser cavity mirrors, 2—holder with condenser lens and 3—holder with MLA.

### 2.3. The Influence of Pump Fluence Distribution on Ti:Sapphire Laser Operation

In spite of the good technical characteristics obtained with homogenization, the influence of pump fluence distribution, w(x, y), on Ti:Sapphire laser operation remained unclear. We describe the pumping spot patterns with different w(x, y) used in our study. Figure 9 shows 2-D patterns and the appropriate w-profiles. Pattern (a) has a structure with relatively slow variations of w. Pattern (b) has a more regular structure with diffraction O-rings at a spatial frequency ~2.5 mm$^{-1}$ in the radial direction. Pattern (c) consists of different size speckles; most of them are less than 0.04 mm. Therefore, the spatial spectrum of this pattern includes frequencies ~25 mm$^{-1}$ and above. MLA pattern (d) consists of horizontal and vertical interference strips with various widths and periods. At the central area of the pattern, spatial frequencies are 17–26 mm$^{-1}$. A bright spot at the center of the patterns (c) and (d) is caused by the part of the light transmitted through the homogenizer without phase modulation ("0"-order). The intensity of this spot can be reduced by the adjustment of the distance between the condenser lens and the Ti:Sapphire crystal.

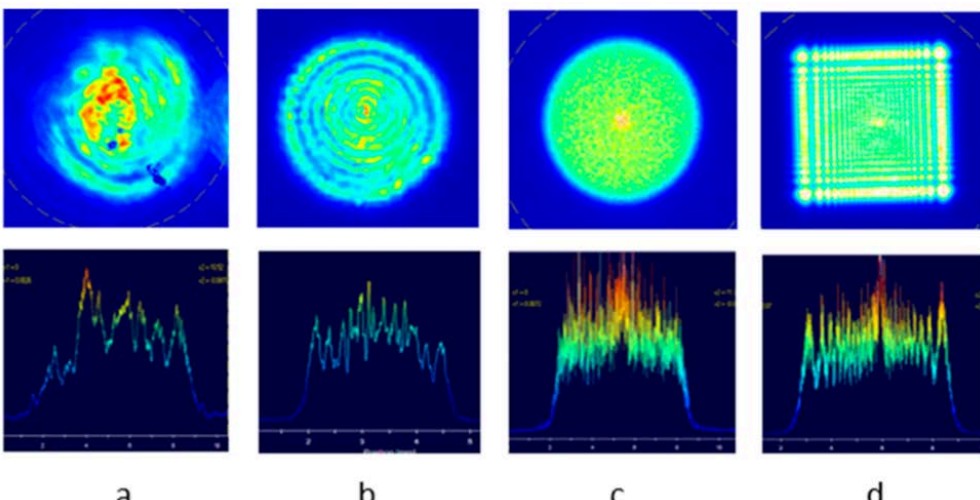

**Figure 9.** Two-dimensional pump beam patterns (upper row) and w-profiles (lower row) of multimode laser (**a**), singlemode laser (**b**), after homogenization of multimode laser by DD (**c**) and by MLA (**d**). Spot size at (**b**) is 3 mm, and at other patterns is 6 mm. Adopted with permission from ref. [17,18]. Copyright 2019 © The Optical Society.

An evident difference between patterns (a) and (b), and (c) and (d), is the high spatial frequency interference modulation of $W_{PUMP}$, caused by homogenizers in the case of pumping by the light with high coherence. In addition, the fluence modulation depth, $\xi = 2\,[w_{(MAX)} - w_{(MIN)}]/[w_{(MAX)} + w_{(MIN)}]$, is of the same order in all of the patterns, but slightly lower in the singlemode laser. Therefore, the differences in local gain would remain even after homogenization. This means that a simple model of multichannel generation with equalized gain cannot completely explain an improvement in temporal characteristics, which we obtained using homogenizers.

In [24], it was asserted that similar interference modulation does not have a significant influence on a laser beam in high-power amplifiers and does not lead to damage of the laser crystal's input surface. Considering generation in the Ti:Sapphire laser, we must keep in mind that there are essential differences between pumping light characteristics and optical scheme configurations used for high-power amplifiers and those used in our laser:

The use of multi-pass scheme configurations in final amplifiers leads to spatial averaging of gain within the amplified beam spot.

Saturation of gain in the final amplifiers reduces the spatial modulation of fluence within the input beam, which is the result of preamplifiers pumped by Nd:YAG lasers.

Indicated factors can reduce the influence of pump light interference on Ti:Sapphire laser beams in an amplifier system, but they did not in our case.

The analysis of pump patterns in Figure 9 lead us to the assumption of an additional mechanism that can produce the coupling and synchronization of the generation within the Ti:Sapphire laser aperture. We suppose that the spatial modulation of pumping light leads not only to gain nonuniformity but also induces the related optical nonuniformities in the crystal due to the refraction index change (RIC) effect [25,26].

Optical nonuniformities should cause the small-angle scattering of laser light. If the representative size of nonuniformities is large (or spatial frequencies are small), like in patterns (a) and (b) in Figure 9, then scattering angles should be small. On the contrary, scattering angles produced by patterns (c) and (d) should be large. We suppose that the scattering of generated light in the Ti:Sapphire laser can lead to synchronization of the generation over large areas of aperture if scattering intensity and angles are sufficiently large.

It is noteworthy that the natural intrinsic scattering in high-quality Ti:Sapphire crystals is very small—the scattering coefficient is less than $0.001\ \mathrm{cm}^{-1}$—which does not influence laser generation. Direct observation of pump-induced scattering is a complicated task because this process is non-stationary and requires high $W_{PUMP}$ for excitation, so it is difficult to separate and register the scattered part of the probe light on the pump light's background. To confirm the existence of pump-induced scattering, we used another approach.

We studied the angular distribution of the Ti:Sapphire laser radiation at different levels of $W_{PUMP}$, using pump patterns with various $w(x, y)$ distribution, shown in Figure 9. It was observed that the angular characteristics of the Ti:Sapphire laser with MLA and DD, under pumping at a high $W_{PUMP}$ level, acquired features associated with intracavity scattering. To explain this, we refer to the results obtained in [27–29], which demonstrate the influence of intracavity scattering on laser generation. Later results [27–29] were summarized in a monograph [30]. A short explanation, based on [30], is provided below.

The initial reason for light scattering is phase modulation $\delta\Psi(x, y)$ obtained by the light wave after transmission through transparent media with small-scale (in the lateral direction) optical nonuniformities. For small $\delta\Psi$, a part of the light, which is scattered after a single pass through nonuniform media, can be calculated as the squared standard deviation $\{StD\,[\delta\Psi(x, y)]\}^2$.

In flat resonators with large $N_F$, even weak light scattering can increase the angular divergence of laser radiation manifold (in our case, $N_F \geq 1000$). The reason for the high sensitivity of such resonators to scattering is due to a very small frequency difference between different order transversal modes. As a result, even weak coupling between such modes leads to the junction of generated light into complexes of modes with the same frequency. Such complexes, which consist of a large number of undistorted flat cavity

high-order transversal modes, with random phase and intensity, are real modes of large $N_F$ flat resonators with intracavity scattering. At a high pump level, these complexes predominate in laser generation, resulting in a large angular divergence of radiation.

The mode structure and angular distribution of laser light depends on the characteristics of scattering: intensity, determined by integral scattering coefficient $\alpha$, and scattering angle $2\vartheta_0$. If $\vartheta_0 \geq (\lambda/L_{EQ})^{0.5}$, where $\lambda$ is the light wavelength and $L_{EQ}$ is the equivalent resonator length, then the far-field radiation distribution, in addition to a central spot, will include O-rings with an angular diameter of $2(\beta\lambda/L_{EQ})^{0.5}$, where $\beta = 1, 2, 3 \ldots$ These O-rings might contain a considerable amount of laser energy, giving rise to the wings of angular distribution.

We should mention that the abovementioned results only reveal some general features of laser angular characteristics. The properties of active media (except scattering), the dynamics of generation and pump characteristics have not been considered here. Nevertheless, these results show the features of laser light angular distribution caused by light scattering, so we can take into consideration the appearance of such features (first, the appearance and growing of the wings) in experimental characteristics as an indicator of intracavity light scattering.

The angular distribution of Ti:Sapphire laser radiation was investigated for three different pump patterns, which are shown in Figure 9b–d. The parameters of Ti:Sapphire laser were the same for all three pump patterns: we set up a laser resonator with flat pump and output mirrors; the reflection coefficient of the output mirror was 70%, the Ti:Sapphire crystal thickness was 3 mm, the distance between mirrors was $6.5 \pm 0.3$ mm, and the pump spot size was ~5 mm.

The results of the experimental investigation are shown in Figures 10–15. Figure 10 shows the angular distribution of the Ti:Sapphire laser energy under single mode pumping at two different W. Here, $\gamma$ is a part of total laser energy, irradiated within angle $\varphi$. Angular divergence, measured at the (1/e) level (63% of total laser energy are concentrated within this angle), was 8 mrad at W = 1 J/cm$^2$ and did not increase with the increase in pump fluence (at least, up to 1.5 J/cm$^2$). The wings of the angular distribution were small; more than 90% of laser energy was concentrated within 12–15 mrad.

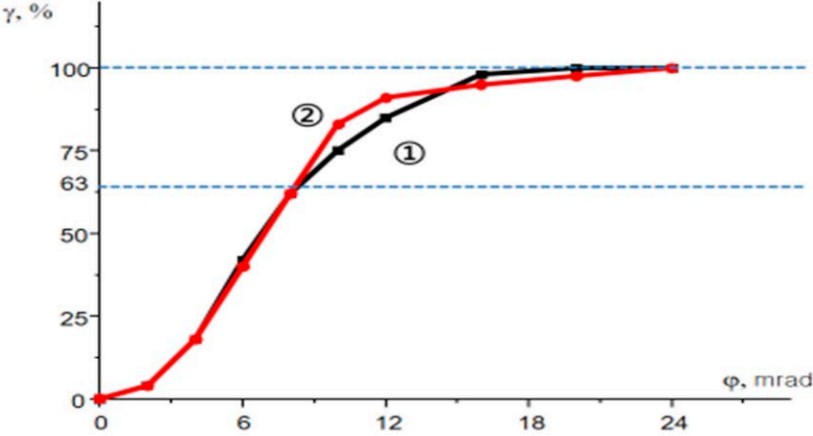

**Figure 10.** The angular distribution of Ti:Sapphire laser energy for single-mode pumping: 1—$W_{PUMP}$ = 1.0 J/cm$^2$; 2—$W_{PUMP}$ = 1.5 J/cm$^2$. Reprinted with permission from ref. [18]. Copyright 2019 © The Optical Society.

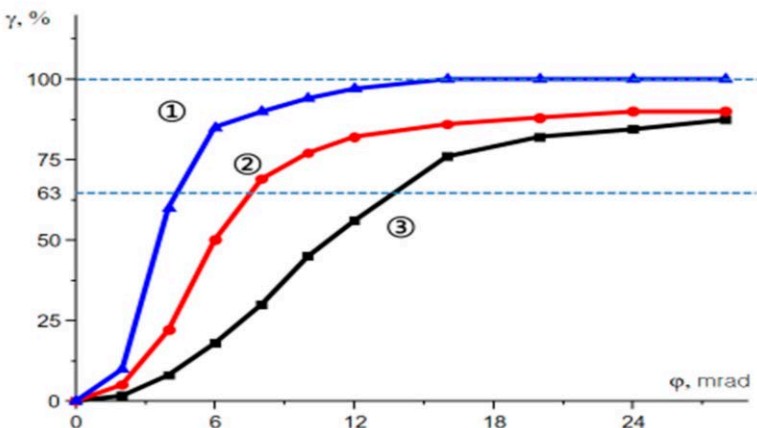

**Figure 11.** The angular distribution of Ti:Sapphire laser energy for DD at: 1—$W_{PUMP} = 0.5$ J/cm$^2$; 2—$W_{PUMP} = 0.75$ J/cm$^2$; 3—$W_{PUMP} = 1.05$ J/cm$^2$. Reprinted with permission from ref. [18]. Copyright 2019 © The Optical Society.

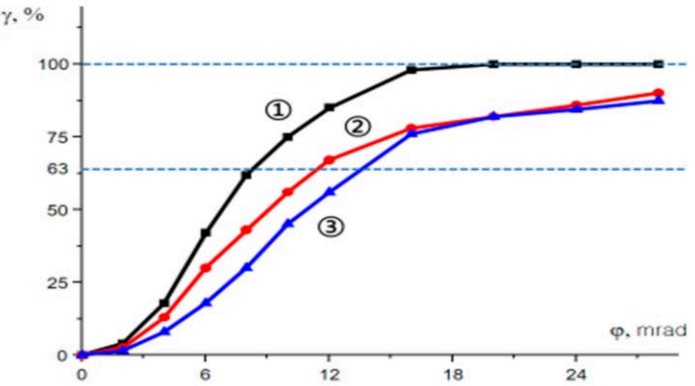

**Figure 12.** The angular distribution of Ti:Sapphire laser at W = 1.05 J/cm$^2$ for different pump patterns: 1—single-mode, 2—MLA, 3—DD. Reprinted with permission from ref. [18]. Copyright 2019 © The Optical Society.

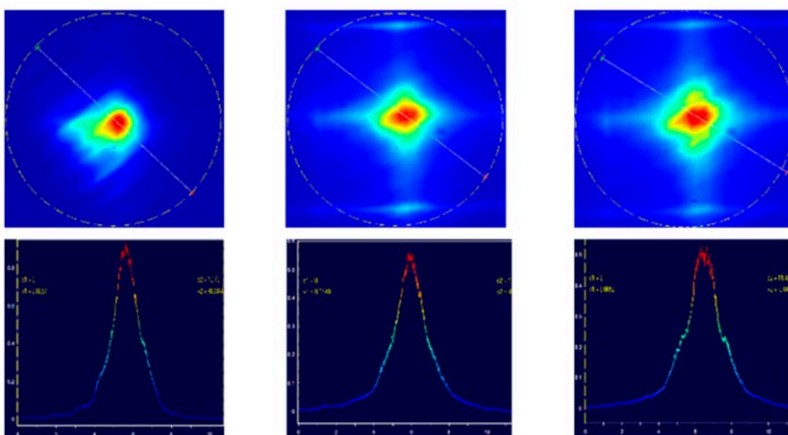

**Figure 13.** Far-field distribution of the Ti:Sapphire laser radiation with MLA pump. Two-dimensional patterns and the appropriate intensity profiles were recorded at the focal plane of 800 mm lens at different levels of pump fluence: left—W = 1 a. u.; middle—W = 1.7 a. u.; right—W = 2.9 a. u. Horizontal scale of the profiles is 1 mm per division. Reprinted with permission from ref. [18]. Copyright 2019 © The Optical Society.

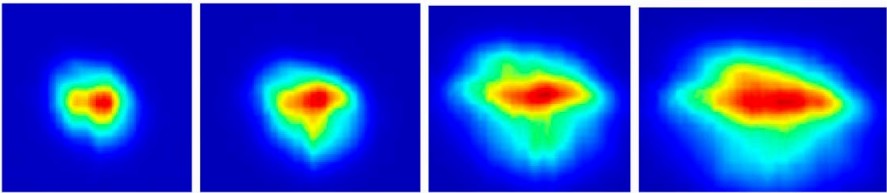

**Figure 14.** Laser far-field distortion with the increase in pump fluence under pumping with one-dimensional MLA. W increases from left to right of the image by 2.9 times. Reprinted with permission from ref. [19]. Copyright 2019 © SPIE.

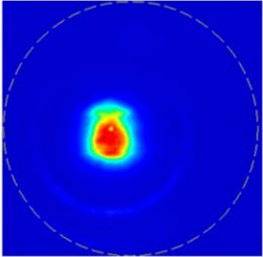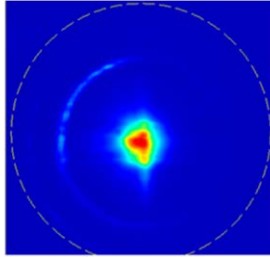

**Figure 15.** O-rings in the far field of Ti:Sapphire laser with the single-mode pump. The right pattern was recorded when AR coatings at the crystal were damaged. Light scattering at damaged areas resulting from increased intensity of O-rings. Reprinted with permission from ref. [18]. Copyright 2019 © The Optical Society.

The divergence, determined as the angular diameter of the laser far-field central spot at the $1/e$ level, was 5.5 mrad, which is smaller than the divergence of 8 mrad determined by 50% energy. In the following, we will indicate only the value of angular divergence, determined by energy, as more relevant to our practical applications.

Another situation was observed with the MLA and DD pump patterns. In both cases, the angular divergence considerably increased with the increase in W. For the DD pump pattern (Figure 11), when W was increased to 1.05 J/cm$^2$, the angular divergence at the $(1/e)$ level increased to 13 mrad, which is much larger than the 8 mrad obtained under the single-mode pump. We should emphasize a rapid growth of the wings of angular distribution here in comparison to the wings observed under the single-mode pump. With DD pumping, when W exceeded 0.5 J/cm$^2$, more than 15% of laser energy was irradiated into angles of larger than 20 mrad.

A comparison of the angular energy distribution at W = 1.05 J/cm$^2$ for all three pump patterns at the $1/e$ level is shown in Figure 12. We observed the minimum angular divergence of 8 mrad for the single-mode pump pattern; a larger divergence for the MLA pump pattern—11 mrad; and the largest divergence for the DD pump pattern—13 mrad. The main difference between angular distributions for MLA and DD pumping consists of the different width of the central spot, which is larger for DD. At the wings, angular distribution for these pump patterns is almost identical. We again emphasize that the appearance of large wings in angular distributions for MLA and DD pumping is the most distinct feature in comparison to angular distribution for single-mode pumping, where the wings are small and almost all energy is concentrated within 15 mrad.

The lateral structure of the pump patterns revealed in the far-field distribution of laser radiation is shown in Figure 13. Single-mode and DD pump patterns are axially symmetric, and the appropriate laser far-field patterns are also axially symmetric. In the case of the MLA pump pattern, which has 4$^{TH}$ order rotational symmetry, the far field also consists of the same symmetry.

The results obtained with 2-D MLA are shown above. In comparison, the far-field pattern, obtained with one-dimensional MLA, consisting of cylinder lenses, while having the same pitch, lens size and focal length as in two-dimensional MLA, is shown in Figure 14. This MLA produced the modulation of W only in the horizontal direction. As expected,

induced scattering in the horizontal direction should be larger than in the vertical direction, so when W increased, the central spot of angular distribution increased mainly in the horizontal direction, and less in the vertical direction, parallel to the cylinder lens axis.

For the MLA and DD pumping, we found that laser light intensity in the axial (normal to resonator mirror surface) direction remained almost unchanged, whereas pump fluence increased 2.9 times, and laser output energy also increased by more than three-fold. All patterns, shown in Figures 13 and 14, were recorded at equal optical attenuation and equal electronic gain of the beam profiler camera. The beam intensity profiles in Figure 13 illustrate the process of laser light energy redistribution from an axial direction to the wings when $W_{PUMP}$ increases. It is worth mentioning that the observed energy redistribution was analyzed and explained theoretically in [29].

Predicted and observed in [27,28], O-rings in the far field intensity distribution (Figure 15) were also observed in our experiments under pumping with each of the three patterns, even with a single-mode pattern. The appearance of O-rings in the last case, in our opinion, might have resulted from a weak light scattering caused by dielectric coatings at Ti:Sapphire crystal and laser resonator mirrors. Damage of the Ti:Sapphire crystal AR coatings led to an increase in light scattering and, finally, an increase in the intensity of O-rings. For single-mode and DD pumping, the O-rings had a round shape, but for MLA pumping, they had a hexagonal shape (not shown here). The angular diameter of the first O-ring, shown in Figure 15, was 18.5 mrad. The calculated angular diameter was 18.95 mrad, which is in a good agreement with the experimental value. We must emphasize that scattering in the case of single-mode pumping was very week and did not lead to the synchronization of the generation.

One can estimate the magnitude of phase shift $[\delta\Psi]_{MAX}$, induced by pump light in the Ti:Sapphire crystal using the data obtained in [25]. It was indicated that the refraction index n changed due to population inversion (electronic component) and heating (thermal component) of the crystal. The total RIC for σ-polarized light at 633 nm was about $6 \times 10^{-24}$ cm$^3$ × N, where N is the concentration of exited Ti$^{3+}$ ions. The thermal component of RIC was determined as ~1/3 of the total value. Our laser operates on π-polarization at 785 nm. According to [26], the electronic component of RIC for this polarization is reduced to zero at ~780 nm, so only the thermal component should be taken into consideration in our case. Supposing low dependence of the thermal component on wavelength and light polarization, in this estimation, we used the value calculated from [25]: RIC (thermal) = $2 \times 10^{-24}$ cm$^3$ × N. The measured gain coefficient, $K_0$, in our Ti:Sapphire crystals was 3 cm$^{-1}$ at $W_{PUMP}$ = 1.5 J/cm$^2$. Using well-known relation: $K_0 = \sigma \times N$, where $\sigma = 4.1 \times 10^{-19}$ cm$^2$ [31] is Ti:Sapphire emission cross section, for $K_0$ = 3 cm$^{-1}$, we obtained N = $7.3 \times 10^{18}$ cm$^{-3}$, and RIC = $1.46 \times 10^{-5}$. Then, we calculated $[\delta\Psi]_{max} = (2\pi/\lambda) \times$ RIC × t = 0.35 rad at λ = 785 nm. Here, t = 3 mm is crystal thickness.

It is possible to estimate the thermal component of the phase shift even more simply, calculating the average heating of the Ti:Sapphire crystal by single pump pulse. Supposing that the heating is caused only by Stokes losses, we obtained the energy dissipated into the heat, $W_{HEAT}$ ~ 0.32 $W_{PUMP}$. Then, supposing 100% absorption of pump energy in the crystal and using data for pure sapphire crystals from Crystran Ltd. (www.crystran.co.uk, accessed on 14 March 2016): density 4 g/cm$^3$, specific heat capacity 0.763 J × g$^{-1}$ × K$^{-1}$, dn/dT = $13 \times 10^{-6}$, for 3 mm thick crystal at $W_{PUMP}$ = 1.5 J/cm$^2$, we obtained $[\delta\Psi]_{AVE}$ = 0.16 rad. The magnitude $[\delta\Psi]_{MAX}$ should be 1.3–1.5 times larger: $[\delta\Psi]_{MAX}$ ~ 0.2–0.24 rad, depending on fluence modulation depth.

To estimate the scattered part of laser radiation in the case of nonuniformities, created by MLA, we supposed the sinusoidal character of phase modulation. Then, we calculated total diffraction losses into both ±1st orders of phase grating with a modulation depth $[\delta\Psi/2]_{max}$ = 0.175 rad, using the relation [32]: $\eta = [J_1(\delta\Psi)]^2$, where η is the efficiency of diffraction into the 1st or -1st order, and $J_1$ is the first-order Bessel function. We obtained 2 η = 1.6% per single pass. It is notable that we believe the structure of w(x, y) is unchangeable within the Ti:Sapphire crystal. In practice, one can observe slow structure variations

when the distance from the condenser lens to the crystal changes. We can neglect these variations (at least for MLA and DD with parameters that produce a pattern with spatial frequencies ~25 mm$^{-1}$) in calculations, because the Ti:Sapphire crystal thickness is much lower than the condenser lens focal length of 60 mm.

For pseudo-random $W_{PUMP}$ distribution, like in the case of DD, we can calculate the relative value of scattered radiation, supposing that $\delta\Psi(x, y)$ is linearly proportional to w(x, y), and calculate the standard deviation StD (w) directly from experimental beam profiles, using beam profiler software. For our DD pump pattern, we obtained StD(w) ~0.29. Then, we calculated the relative value of scattered radiation: $\{StD [\delta\Psi(x, y)]\}^2 = \{[\delta\Psi(x, y)]_{max} \times StD (W)\}^2 = 2.3\%$. Expressing this value in standard form, we obtained an integral scattering coefficient $\alpha = 0.077$ cm$^{-1}$, related to a gain coefficient $K_0 = 3$ cm$^{-1}$.

The scattering coefficients ($\alpha$) for the comparison of several materials are presented in Table 3:

**Table 3.** Scattering coefficients of optical materials. Adopted with permission from ref. [18]. Copyright 2019 © The Optical Society.

| Material | $\alpha$, Intrinsic, cm$^{-1}$ | Ref. |
|---|---|---|
| Ruby and Ti:Sapphire | 0.001 | [29] |
| Nd$^{3+}$:YAG crystal | 0.002 | [29] |
| Nd$^{3+}$:YAG ceramic | 0.004 | [33] |
| **Material** | **$\alpha$, induced, cm$^{-1}$** | |
| Ti:Sapphire with MLA | 0.053 | Calc. |
| Ti:Sapphire with DD | 0.077 | Calc. |

One can see that the induced scattering coefficient here is larger than the appropriate coefficients of the intrinsic scattering of indicated optical materials. Therefore, it is not surprising that the influence of pump light spatial modulation on Ti:Sapphire laser generation is considerable.

We can conclude the following:

1. The induced scattering in our application caused a positive effect, leading to the synchronization of laser generation over aperture. This allowed us to obtain stable subnanosecond pulses from the Ti:Sapphire laser under nanosecond pumping and increase pulse energy in comparison to subnanosecond pumping. Scattering intensity <1–2% is sufficient for synchronization if the scattering angles are large enough. Angles larger than ~30 mrad (estimated from our pump patterns) are sufficient; angles < 5 mrad are definitely insufficient.
2. The concomitant increase in the angular divergence to 15–20 mrad in our application was not critical. More than 90% of the laser energy can be delivered to the target using a standard articulated arm. It is worth mentioning that small-angle scattering did not reduce the energy efficiency in the lasers with large $N_F$ because the scattered light was not lost, but still participated in the generation process.
3. The induced scattering, which is an additional mechanism for the synchronization, of course, cannot compensate for big differences in w(x, y) at low spatial frequencies. The parameters of the homogenizers must be properly selected to eliminate such differences.

Hypothetically, it is possible to obtain synchronous laser generation in such laser without the assistance of scattering in the case of flat-top w(x, y). However, a similar distribution in practice can be realized only with pump sources of lower coherence, for example, with Nd:glass lasers.

It is notable that small-angle scattering for synchronization in the Ti:Sapphire laser can also be produced, for example, by the deposition of special coatings on the surface of a laser

cavity. However, such technical solution seems unreasonable, because, as a rule, nonuniformities in the coatings considerably reduce their LIDT, which is not acceptable here.

## 3. Conclusions

The results described and discussed in this review show that active Ti:Sapphire crystals can be successfully used in applications other than laboratory equipment for scientific research. The novel engineering solutions make it possible to develop the devices, based on Ti:Sapphire lasers with characteristics, required for several medical applications. Such devices can possess technical and economic advantages in comparison to the analogs represented in the medical market.

Most of the experimental results shown here were obtained before 2020. At present, we continue to improve the characteristics of Ti:Sapphire lasers. In particular, we completed the development of a new model, PALLAS-2, with a pulse repetition rate that increased from 100 to 300 Hz, and we are now preparing this model for serial production.

Such process as the appearance of the induced scattering in Ti:Sapphire crystals, nevertheless, has not yet been confirmed by indirect evidence. Therefore, more detailed experimental study of this process via methods similar to that used in [34] for Nd:YAG crystals is appropriate.

**Author Contributions:** A.T. and H.C. contributed equally to this work. All authors have read and agreed to the published version of the manuscript.

**Funding:** This research received no external funding.

**Institutional Review Board Statement:** Not applicable.

**Informed Consent Statement:** Not applicable.

**Data Availability Statement:** Not applicable.

**Conflicts of Interest:** The authors declare no conflict of interest.

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
