# Peer review of "Engineering of Ti:Sapphire Lasers for Dermatology and Aesthetic Medicine"

_applsci, doi:10.3390/app112210539_

Round 1

Reviewer 1 Report

This manuscript provides a review of the new engineering solutions for Ti: Sapphire lasers, obtained at Laseroptek during the development of laser devices for dermatology and aesthetic medicine. Two medical laser devices, PALLAS and HELIOS-4, which include Ti: Sapphire lasers in combination with other modules such as Nd: YAG lasers, nonlinear frequency converters, etc. have been considered.

The review is useful and well written.  It is offering basic achievements on the subject. It can be published after several changes.

1). The quality of figures 3, 4, 10 (right) has to be improved.

2). The colorbar has to be added to figures 10 (left) 12, 16, 17, 18.

3). The number of abbreviations has to be decreased in order to simplify the reading of the paper.

4). In Conclusion after the sentence “Such devices can possess the technical and economic advantages in comparison with the analogs, represented at the medical market.” several sentences have to be added. They have to be devoted to the description of these technical and economic advantages.

Author Response

Thank you for reviewing of our paper.  We will response following to your numbering.

  1. It was done in Introduction. There we explain possible advantages of application of Ti: Sapphire lasers in laser medical systems instead excimer lasers and PICOSURE more detail. Conclusion at the end of the review only summarizes briefly this information.
  2. We use only standard abbreviations, common for laser-related journals, and explain them when use for the first time. Most of optics-laser-physics- related readers are familiar with these abbreviations.

1 and 2. We agree that the quality of the figures could be better and the colorbar, in general, is useful. Nevertheless, please take in mind that these figures already have been accepted and published at our papers by the appropriate journals. We only reprint them in our review. So we believe it is possible to use them at our review in the original form.

Reviewer 2 Report

In the manuscript ' Engineering of Ti: Sapphire lasers for dermatology and aesthetic medicine, ' the authors Tarasov et al reviewed their products of Ti: Sapphire lasers applied in the medical fields considering some engineering solutions such as pulsed performing, beam quality and system designing. Personally, the authors just finished a quite well presented introduction of a kind of product with some engineering preference. The manuscript is suggested to be accepted after the following questions have been answered.

(1) As the laser systems are regarded as the products, the authors are encouraged to introduce the cooling designing such as the preferred chiller parameter and temperature stability in time domain.

(2) The authors are encouraged to compare their laser system with the similar products of other vendors in a certain or serval aspects such as the beam quality under the similar pulse energy

Author Response

Thank you for the reviewing of our manusctipt. This special issue is devoted to Ti: Sapphire lasers. We emphasize (see Introduction at our manuscript) that our main goal is NOT the review and  comparison of the characteristics of different medical laser systems, which are shown at the market (they do not include Ti: Sapphire lasers). We only review some of our original solutions for Ti: Sapphire lasers, which allow to make medical laser systems , based on such lasers, less expensive and more simple, than other systems, represented at the market. And we do not consider other parts , included into our systems, like Nd: YAG laser, chiller, etc.

So we beleive that your suggestions are out of score of out paper.